# SignFlow Bipartite Subgraph Network For Large-Scale Graph Link Sign Prediction

**Yixiao Zhou**
Wangxuan Institute of Computer Technology
Peking University

**Hongxiang Lin**
Wangxuan Institute of Computer Technology
Peking University

**Huiying Hu**
Wangxuan Institute of Computer Technology
Peking University

**Tuo Wang**
Wangxuan Institute of Computer Technology
Peking University

**Xiaoqing Lyu**[*]
Wangxuan Institute of Computer Technology,
State Key Laboratory of Multimedia Information Processing
Peking University

## Abstract

Link sign prediction in signed bipartite graphs, which are extensively utilized across diverse domains such as social networks and recommendation systems, has recently emerged as a pivotal challenge. However, significant space and time complexities associated with the scalability of bipartite graphs pose substantial challenges, particularly in large-scale environments. To address these issues, this paper introduces the SignFlow Bipartite Subgraph Network (SBSN), balancing sublinear training memory growth through a heuristic subgraph extraction method integrated with a novel message passing module, with optimal inference efficiency achieved via the node feature distillation module.

Our subgraph sampling approach reduces the graph size by focusing on neighborhoods around target links and employs an optimized directed message passing mechanism to aggregate critical structural patterns. This mechanism allows SBSN to efficiently learn rich local structural patterns essential for accurate sign prediction. Furthermore, to overcome the inefficiency of subgraph sampling-based models during inference, SBSN incorporates a node feature distillation module after the first training stage. This module distills subgraph features into node features, enabling fast inference while retaining the rich structural information of subgraphs.

Experiments reveal that SBSN shows superior performance in both medium- and large-scale datasets, efficiently managing memory and computational resources, making it a scalable solution for extensive applications. The implementation of SBSN is publicly available at `https://github.com/WICTSA/SBSN`.

---

[*]Correspondence: lvxiaoqing@pku.edu.cn. Author emails: chnzyx@pku.edu.cn, linhongxiang@stu.pku.edu.cn, hhy2024@pku.edu.cn, tuowang25@stu.pku.edu.cn, lvxiaoqing@pku.edu.cn

39th Conference on Neural Information Processing Systems (NeurIPS 2025).

# 1 Introduction

In the digital society nowadays, the diversity of evaluations among individuals is prevalent and manifests significant complexity. This phenomenon spans various domains such as social network interactions [4, 17, 29], personalized feedback in recommendation systems [19, 20, 21], and peer review in academic research [9, 32]. To fully analyze such social phenomena, researchers frequently employ the signed bipartite graph model. In a signed bipartite graph, nodes are partitioned into two disjoint sets, e.g., users and commodities, or reviewers and papers. Links between these sets are represented by edges with positive or negative signs, where positive edges signify favorable connections or ratings and negative edges denote unfavorable ones. Predicting the signs of these links, known as the link sign prediction task, is a fundamental problem in signed bipartite graph research. Existing methods can generally be categorized into two types: feature-based and subgraph-based approaches.

Despite progress in the field [24, 25, 9, 32, 6], link sign prediction in signed bipartite graphs remains challenging. First, feature-based approaches commonly leverage balance theory to model node interactions. However, such methods often suffer from growth in space complexity as the size of the dataset increases. Furthermore, balance theory cannot model all signed graph formation patterns[18], leading to substantial information loss. Second, subgraph-based approaches have recently emerged, extracting and encoding subgraphs for each query during training and inference. Although this strategy offers local structural advantages, it faces limitations in scalability and inference speed, which significantly restrict its applicability in large-scale scenarios.

These limitations become even more pronounced in real-world scenarios, such as user preferences on e-commerce platforms and user ratings on video streaming sites, where both the number of nodes in each set and the number of edges grow to massive scales. Performing message passing and feature learning directly on the entire bipartite graph becomes infeasible, resulting in prohibitive computational costs. The scalability of current models is thus severely hindered by their high resource demands, limiting their applicability in practical, large-scale settings.

To address these challenges, we propose a link sign prediction method, the SignFlow Bipartite Subgraph Network (SBSN), to accomplish the prediction task:

1. **Scalable Subgraph Training**: To tackle the rapid growth of space complexities with increasing dataset size, we transform global node feature training into local subgraph feature training. Unlike traditional methods that sample k-hop neighbors of nodes $u$ and $v$ separately, our subgraph sampling ensures each node lies on a simple path connecting $u$ and $v$, greatly improving information efficiency. By leveraging the properties of bipartite graphs, our subgraph sampling strategy establishes a topological structure to facilitate subsequent subgraph feature learning. Such a subgraph training design ensures that the space complexity of our model scales sublinearly with the size of the dataset.

2. **Directed SignFlow Passing**: To effectively capture the structural information of subgraphs, we propose the SignFlow Aggregator within SBSN. This module aggregates all path information connecting source nodes and target nodes within the topological subgraph, enabling comprehensive structural representation. The edge-feature-based SignFlow Aggregator further enriches the subgraph representation, surpassing node-feature-based methods such as GCN in information richness.

3. **Feature Distillation for Efficient Inference**: To overcome the inference inefficiencies of subgraph-based models compared to feature-based models, we introduce a node feature distillation module. This additional training step distills subgraph features into node features, allowing our model to achieve better inference efficiency. Also, node features serve a broad range of downstream tasks, including node clustering and node classification.

4. **Theoretical and Empirical Complexity Analysis**: We provide both theoretical and empirical analyses of the model's complexity, demonstrating how computational and memory costs scale with dataset size $E$. Extensive experiments on medium- and large-scale datasets further validate the scalability and accuracy of our approach.

## 2 Related Works

### 2.1 Signed Graph Representation Learning

Signed graphs have garnered significant attention due to the rapid growth of social networks[4, 17, 29] and recommender systems[19, 20, 21]. Numerous tasks in signed graph analysis have been explored, such as node classification[26], node ranking[23], link sign prediction[9, 32], community detection[2], and visualization[28]. Among these tasks, link sign prediction is particularly important. Signed Graph Representation Learning (SGRL) is an effective approach to analyze the complex patterns in real-world signed graphs with the co-existence of positive and negative links[33]. Early methods for learning representations of signed graphs were based on random walks[12, 30, 13] and matrix factorization[3, 15]. In recent years, deep learning has been applied to signed representation learning. SiNE[28] extracts structural information from triangle motifs and designs an objective function based on balance theory[8]. SGCN[24] became the first signed graph neural network model, extending GCN[14] and using balance theory to determine positive and negative relationships between nodes in multi-hop neighborhoods. Similarly, models like SiGAT[10], SNEA[16], and SDGNN[11] use graph attention networks to learn signed graph representations. SBGCL[32] proposes a contrastive learning method for robust signed graph representation learning. SGA[31] proposed a novel data augmentation approach tailored for SGNNs. By introducing a new perspective on data augmentation, SGA aims to significantly enhance the training process.

However, SGRL methods typically perform message passing through GNN layers across the entire bipartite graph, which makes GPU memory management challenging. As the dataset size increases linearly, the GPU memory consumption of such models increases at least linearly. Higher precision message-passing layers often require more memory, making the training of such models on large-scale datasets impractical due to the prohibitive GPU memory consumption. Additionally, balance theory, which is commonly used by SGRL methods, cannot model all signed graph formation patterns[18].

### 2.2 Subgraph Embeddings and Prediction

Subgraph-based methods have also been extensively explored for graphs. SUBGNN [1] introduces a subgraph neural network with a novel routing mechanism to learn disentangled subgraph representations. SELO [6] extracts enclosing subgraphs for each target pair of nodes and encodes these subgraphs into vectors through a unique linear optimization (LO)-based approach. SELO introduces a likelihood matrix representation for subgraphs, which captures triad and high-order cycle patterns rather than pairwise interactions.

However, SELO does not strictly limit subgraph size during the matrix computation phase, leading to large-scale matrix operations on large datasets. Moreover, the above subgraph learning algorithms require subgraph sampling for each query during the inference phase, resulting in suboptimal inference performance compared to SGRL algorithms. Furthermore, subgraph-based methods often lack node features, as they rely on subgraph embedding for inference, which limits their ability to learn rich node representations and makes them less applicable to other node-based downstream tasks.

## 3 Problem Formulation of Signed Bipartite Graph Link Sign Prediction

A signed bipartite graph is defined as $G = (U, V, E^+, E^-)$, where the two sets of nodes $U = \{u_1, u_2, \ldots, u_{|U|}\}$ and $V = \{v_1, v_2, \ldots, v_{|V|}\}$ are disjoint, $E^+ \subseteq U \times V$ and $E^- \subseteq U \times V$ are the positive and negative edge sets, respectively. Note that $E^+ \cap E^- = \emptyset$ and the two endpoints of any edge in $E^+ \cup E^-$ must come from different sets $U$ and $V$. For example, in Figure 1, the set $U$ could be the set of users in an e-commerce platform, and the set $V$ could be the set of products. A positive edge $(u, v) \in E^+$ represents that the user $u$ rates the product $v$ favorably, and a negative

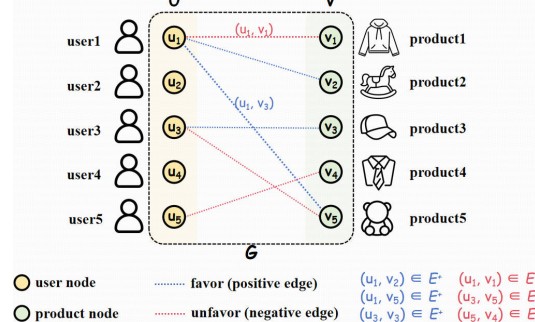

Figure 1: Signed bipartite graph in an e-commerce scenario

edge $(u, v) \in E^-$ represents that $u$ rates $v$ unfavorably. In this paper, we ignore the directions of the edges and treat the graph $G$ as an undirected graph.

Given $G = (U, V, E^+, E^-)$, the goal is to predict the sign of an edge $(u, v) \in U \times V$ that is not in the observed set of edges $E^+ \cup E^-$. Specifically, we aim to learn a function $f$ that, for any pair of nodes $u_i \in U$ and $v_j \in V$, predicts whether the edge between them should be positive or negative. The function $f$ should be able to accurately determine the sign of the link based on the structural properties of the graph and the features of the nodes.

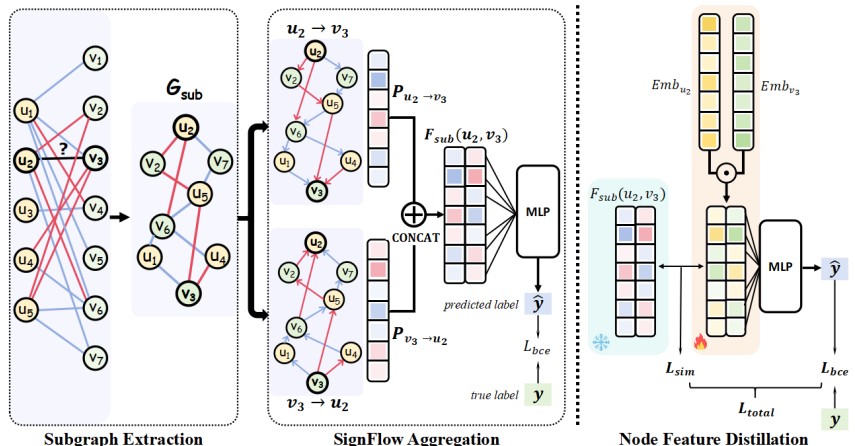

Figure 2: Overview of the SignFlow Bipartite Subgraph Network (SBSN). SBSN comprises three modules: (1) **Topological Subgraph Extractor**, which identifies subgraph $G_{\text{sub}}(u, v)$ for a queried edge $(u, v)$; (2) **Directed SignFlow Aggregator**, which performs edge-based information propagation and weighted aggregation; and (3) **Node Feature Distillation Module**, which enhances inference efficiency in the second training stage.

# 4    Proposed Method

To address the challenges of high computational cost and scalability in large-scale bipartite graphs, we propose the SignFlow Bipartite Subgraph Network (SBSN), as illustrated in Figure 2. Our model comprises three main modules: a topological subgraph extractor (Section 4.1), a directed SignFlow Aggregator (Section 4.2), and a node feature distillation module (Section 4.5). The first two modules are used for the primary link sign prediction training. Specifically, the topological subgraph extractor identifies the subgraph for a queried edge, while the SignFlow Aggregator performs edge-based information propagation and weighted aggregation to generate subgraph embeddings. These embeddings are fused and passed through an MLP to predict the link sign based on a weighted binary cross-entropy loss. The third module, the Node Feature Distillation Module, is employed during the second stage of training to reduce resource consumption during inference and to obtain node features applicable to a broader range of downstream tasks.

For a queried edge $(u, v)$, the model employs a Topological Subgraph Extractor. This extractor identifies a topological subgraph $G_{\text{sub}}(u, v)$ that is highly relevant to the queried edge. The subgraph contains multiple paths with $u$ and $v$ as endpoints, facilitating subgraph structure embedding and inference.

Once the subgraph structure is obtained, directed information propagation is performed. The SignFlow Aggregator aggregates path information at the edge granularity, allowing nodes in subsequent layers to aggregate features from all incoming edges. This module also derives attention values for each neighbor in the previous layer based on edge features and then performs weighted aggregation.

During the training phase, we use the aggregated subgraph encoding to predict the sign of the queried edge and compute the loss function. In the inference phase, subgraphs are directly extracted for

prediction. Alternatively, with additional training of the node feature distillation module, we achieve significant improvements in inference speed at the cost of minimal performance loss.

## 4.1 Topological Subgraph Extractor

As illustrated in Figure 3, the topological subgraph extractor constructs a subgraph iteratively with $u$ and $v$ as the first and last layers, respectively. This ensures that the subgraph effectively captures the structural information of the bipartite graph while maintaining computational efficiency. The detailed process is described below:

### 4.1.1 Iterative Node Sampling for Subgraph Construction

Given a queried edge $(u, v)$ in a bipartite graph $G = (U, V, E^+, E^-)$, the goal is to construct a topological subgraph $G_{\text{sub}}(u, v)$ with $u$ and $v$ as the source and target nodes. The subgraph is expanded iteratively by sampling intermediate layers of nodes from $U$ and $V$.

Let $G_{\text{sub}}^{(0)} = \{u, v\}$ represent the initial subgraph, where $u$ and $v$ are the only nodes. During each iteration $t$, two sets of nodes, $L_t^u \subseteq U$ and $L_t^v \subseteq V$, are sampled as intermediate layers. The process is as follows:

1. **Neighbor Node Selection:** Define the neighbors of a node set $L$ in $U$ or $V$ as:
$$N(L) = \{w \mid \exists (x, w) \in E, x \in L\}.$$
   At iteration $t$, the candidate sets for new nodes are:
$$N(L_t^u) \cup N(L_t^v),$$
   where $L_t^u$ and $L_t^v$ are the current intermediate layers of nodes in $U$ and $V$, respectively.

2. **Sampling New Nodes:** New nodes for the subsequent layers, $L_{t+1}^U$ and $L_{t+1}^V$, are sampled from their respective candidate sets using one of several heuristic strategies: *Random Sampling* (uniform selection), *High-Degree Sampling* (prioritizing nodes with the highest degrees), or *Low-Degree Sampling* (prioritizing nodes with the lowest degrees).
   The sampled nodes must satisfy:
$$L_{t+1}^U \cap G_{\text{sub}}^t = \emptyset, \quad L_{t+1}^V \cap G_{\text{sub}}^t = \emptyset,$$
   ensuring that no duplicate nodes are added to the subgraph.

3. **Subgraph Update:** Update the subgraph with the newly sampled nodes:
$$G_{\text{sub}}^{t+1} = G_{\text{sub}}^t \cup L_{t+1}^U \cup L_{t+1}^V.$$

The process repeats until the total number of nodes in $G_{\text{sub}}(u, v)$ reaches a pre-defined limit $T_{\text{sub}}$.

**Theorem 1(Optimal Number of Topological Layers).** Let $G$ be a random bipartite graph on $T$ nodes with edge probability $p$. Under a budget of $T$ total nodes, the number of topological subgraph layers $K$ that maximizes the expected number of distinct paths from the first to the last layer is
$$K_{opt} = \max\left\{2, \left\lfloor \frac{Tp}{e} \right\rfloor\right\} + 2.$$
For the full derivation of this result, see Appendix I.

### 4.1.2 Edge Extraction for the Topological Subgraph

After determining the node sets $G_{\text{sub}}(u, v)$, the edges in the subgraph are extracted from the original bipartite graph $G$. The edge set $E_{\text{sub}}$ of the topological subgraph is defined as:

- Include edges $(x, y)$ in $E_{\text{sub}}$ if $x, y \in G_{\text{sub}}(u, v)$ and $(x, y) \in E$ in the original graph.
- Exclude the direct edge $(u, v)$ from $E_{\text{sub}}$, even if it exists in $E$.

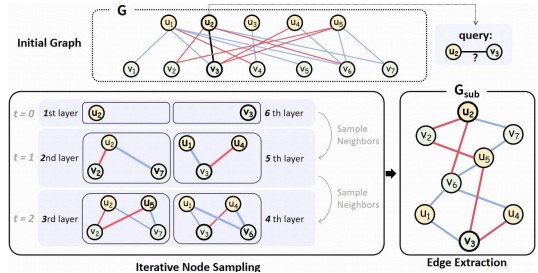

Figure 3: Illustration of the Topological Subgraph Extraction Process

Thus, the edge set is formally expressed as:
$E_{\text{sub}} = \{(x, y) \mid x, y \in G_{\text{sub}}, (x, y) \in E \backslash \{(u, v)\}\}.$

## 4.2 Directed SignFlow Passing

Before training, each node is assigned an initial feature vector sampled from a normal distribution. Let $\mathbf{Z}^U$ and $\mathbf{Z}^V$ represent the embeddings of nodes on the two sides of the bipartite graph, where $\mathbf{Z}_u$ represents the feature of node $u$. During training, we first extract a multi-layer topological subgraph $G_{\text{sub}}(u, v)$ for the query edge $(u, v)$ using the topological subgraph extractor. Suppose $G_{\text{sub}}(u, v)$ consists of $k$ layers; then, the first layer only contains node $u$, and the $k$-th layer only contains node $v$.

Let $\mathbf{Z}_i^{\text{sub}}$ represent the feature matrix of the nodes of $i$-th layer in the subgraph, and let $\mathbf{E}_{i,j}^{\text{sub}}$ denote the edge weight matrix between the $i$-th and $j$-th layers. We use a message passing (MP) function to propagate and aggregate information across layers. The initial feature of the first layer is $\mathbf{H}_1 = \mathbf{Z}_u$.

For the $i$-th layer, the aggregated features $\mathbf{H}_i$ are obtained by fusing the output of the message passing operations from all preceding layers as follows:

$$\mathbf{H}_i = \Phi\left(\{\text{MP}(\mathbf{H}_j, \mathbf{Z}_i^{\text{sub}}, \mathbf{E}_{j,i}^{\text{sub}}) \mid j = 1, 2, \ldots, i - 1\}\right),$$

where $\Phi$ represents the feature fusion mechanism.

The message passing and feature fusion process continues iteratively until the final $k$-th layer. After completing message passing, the feature $\mathbf{H}_k$ at the $k$-th layer is used as the subgraph feature $\mathbf{P}_{u \to v}$ for the multi-layer topological subgraph $G_{\text{sub}}(u, v)$:

$$\mathbf{P}_{u \to v} = \mathbf{H}_k.$$

The message propagation process is referred to as SignFlow Passing in this work because it explicitly incorporates directed information flow across layers, reflecting the directionality of information transfer from node $u$ to node $v$.

Let $N$ be the number of nodes, $T$ the subgraph node budget, $d$ the feature dimension, $L$ the number of message-passing layers, and $B$ the batch size. Encoding an extracted subgraph is dominated by edge-wise message passing and feature transforms; in the worst case the time cost for a single subgraph encoding scales as $O(L\,T^2\,d^2)$, and the peak GPU memory is dominated by persistent node features plus temporary working memory for batch subgraphs, which can be written as $O(Nd + B\,T^2\,d^2)$; importantly, when Node Feature Distillation is used at inference the batch-dependent term related to $T$ disappears and the active computation reduces to $O(B\,d^2)$. Notably, the time to encode one subgraph is independent of the total number of edges $E$, while peak GPU memory scales sublinearly with the dataset size $E$ because only $N$ appears in the persistent memory component (under constant graph density, $E \propto N^2$). Full derivations, constant factors and empirical measurements are provided in Appendix A.

## 4.3 SignFlow Aggregator

To address the issue of information loss due to aggregation solely based on node features, we adopt SignFlow Aggregator to extract and refine edge features, enabling pre-passage of messages through edge features and reducing path information loss, thereby achieving more robust feature aggregation.

In our implementation, the message passing module aggregates features by applying the SignFlow Aggregator separately to the positive and negative edges between two layers of nodes. The results of these two computations are then combined using a feature fusion function. Here, the SignFlow Aggregator is responsible for processing the edge information and node features to generate refined edge features, while the positive and negative edges are handled independently to account for their distinct contributions to the overall aggregation.

### 4.3.1 Node to Edge Feature Transformation

Node features from two consecutive layers are transformed into edge features. Let $\mathbf{H}^U$ and $\mathbf{H}^V$ represent the features of the nodes in the source and target layers, respectively. For each edge connecting nodes $u$ and $v$, the edge feature $\mathbf{H}_{u,v}^E$ is computed as:

$$\mathbf{H}_{u,v}^E = \text{MLP}\left(\text{Concat}(\mathbf{H}^U(u), \mathbf{H}^V(v))\right),$$

where $\mathbf{H}^U(u)$ and $\mathbf{H}^V(v)$ are the features of nodes $u$ and $v$, respectively, and MLP is a multi-layer perceptron.

### 4.3.2 Edge Feature Message Passing

Message passing is then performed on the edge features. For each edge $(u, v)$, where $v$ has a set of neighbors $N_v$ excluding $u$, the edge feature $\mathbf{H}_{u,v}^E$ is updated using the features of edges connecting $v$ with its neighbors $p \in N_v \setminus \{u\}$. The updated edge feature is:

$$\mathbf{H}_{u,v}'^E = \text{Aggregate}\big(\mathbf{H}_{u,v}^E, \{\mathbf{H}_{p,v}^E \mid p \in N_v \setminus \{u\}\}\big).$$

The aggregation is performed by pooling the edge features of the set:

$$\mathbf{H}_{u,v}'^E = \text{Aggregate}\big(\mathbf{H}_{u,v}^E, \text{Pooling}(\{\mathbf{H}_{p,v}^E \mid p \in N_v\})\big).$$

### 4.3.3 Edge Attention Values

The updated edge features $\mathbf{H}'^E$ are reduced to scalar values to compute attention weights $\alpha_{u,v}$ for each edge. For each node $v$ in the target layer, the attention weights are normalized using a softmax function:

$$\alpha_{u,v} = \frac{\exp(\text{Score}(\mathbf{H}_{u,v}'^E))}{\sum_{p \in N_v} \exp(\text{Score}(\mathbf{H}_{p,v}'^E))},$$

where $\text{Score}(\cdot)$ computes a relevance score for the edge feature. In our model, we implement $\text{Score}(\cdot)$ using a linear layer. These attention weights are used to perform weighted aggregation of node features from the source layer:

$$\text{Agg}(v) = \sum_{p \in N_v} \alpha_{p,v} \cdot \mathbf{H}^U(p).$$

### 4.3.4 Edge Feature Integration

Finally, the aggregated node features are combined with the pooled edge features. For each node $v$ in the target layer, the pooled edge features are integrated into the final feature as follows:

$$\mathbf{H}^V(v) = \text{Agg}^h(v) + \text{MLP}\big(\text{Pooling}(\{\mathbf{H}_{p,v}^E \mid p \in N_v\})\big).$$

This formulation ensures that both node and edge information are deeply integrated, enhancing the representation power of the SignFlow Aggregator.

### 4.4 Loss Function

After performing message passing in both directions, we obtain subgraph embedding $\mathbf{P}_{u \to v}$ and $\mathbf{P}_{v \to u}$. These are used to form the fused feature as follows:

$$F_{sub}(u, v) = \text{Concat}(\mathbf{P}_{u \to v}, \mathbf{P}_{v \to u}) \tag{1}$$

The fused feature is then passed through an MLP to reduce its dimensionality to 1, followed by a sigmoid activation. The output is compared with the true label using a weighted binary cross-entropy loss, where the weight is the proportion of positive labels in the batch.

$$\mathcal{L}_{\text{bce}} = -\frac{1}{N} \sum_{i=1}^{N} w_i \left(y_i \log(\hat{y}_i) + (1 - y_i) \log(1 - \hat{y}_i)\right) \tag{2}$$

where $N$ is the number of samples, $y_i$ is the true label, $\hat{y}_i$ is the predicted label, and $w_i$ is the weight for sample $i$. The weight $w_i$ is determined by the proportion of positive labels in the batch.

### 4.5 Node Feature Distillation Module

To achieve a lightweight inference process, we propose a node feature distillation module that approximates the features of subgraphs using the embeddings of nodes $u$ and $v$. Specifically, the edge embedding $Z_{u,v}$ is computed as a linear transformation of the element-wise product of $\mathbf{Z}_u$ and $\mathbf{Z}_v$:

$$\mathbf{Z}_{u,v} = \text{Linear}(\mathbf{Z}_u \odot \mathbf{Z}_v),$$

where $\odot$ denotes the element-wise product. This operation captures interactions between individual dimensions of $\mathbf{Z}_u$ and $\mathbf{Z}_v$, enabling a meaningful representation of their combined information.

To guide $\mathbf{Z}_{u,v}$ in approximating the topological subgraph features, we use a distillation loss defined as the similarity between $\mathbf{Z}_{u,v}$ and $\text{F}_{\text{sub}}(u,v)$, the feature representation of the subgraph obtained from the SignFlow module. We denote this similarity loss as $\mathcal{L}_{\text{sim}}$:

$$\mathcal{L}_{\text{sim}} = \cos(\mathbf{Z}_{u,v}, \text{F}_{\text{sub}}(u,v)).$$

This module is trained after completing the primary training of the SignFlow Aggregator, focusing on further optimizing $\mathbf{Z}_{u,v}$ to enhance inference efficiency with minimal performance loss.

For classification, $\mathbf{Z}_{u,v}$ is passed through a MLP to reduce its dimensionality to one, followed by a sigmoid activation to produce the predicted probability of the edge sign. The output is compared with the true label using a weighted binary cross-entropy loss, denoted as $\mathcal{L}_{\text{bce}}$.

The total loss for the node feature distillation module combines the distillation loss and the binary cross-entropy loss:

$$\mathcal{L}_{\text{total}} = \mathcal{L}_{\text{sim}} + \mathcal{L}_{\text{bce}}.$$

Upon completing the node feature distillation process, inference for a queried edge $u,v$ becomes significantly simpler. Instead of performing complex topological subgraph sampling and message passing, the prediction is directly obtained by computing the element-wise product of $\mathbf{Z}_u$ and $\mathbf{Z}_v$, followed by a MLP. This streamlined approach greatly accelerates the inference process while maintaining high prediction accuracy.

## 5 Experiments

In this section, we introduce seven real-world datasets and baseline models, along with our experimental setup. We then present the average metrics, demonstrating the performance of our model, after which we discuss the results in memory management and conduct ablation studies and parameter studies to evaluate the scalability of our model on large-scale datasets.

### 5.1 Datasets

We conduct a series of experiments on four small-scale and medium-scale real-world datasets: Bonanza[2], U.S. House[5], ML-1M[3] and Amazon-Book[4], and three large-scale datasets: ML-10M[2], ML-32M[2] and Amazon-Book-51M[5]. These datasets cover the academic, social, and political domains, providing us with rich information to validate the performance and applicability of our model. More details about these datasets can be found in Appendix B.

### 5.2 Baselines and Experiment Setting

We evaluate our method, SBSN, against several baselines, including node feature-based methods, Signed GNNs, and Subgraph-based Networks. During the inference phase, for each queried pair of nodes, SBSN extracts a subgraph and predicts the sign using the subgraph embedding, whereas the derived SBSN-node directly leverages the node features distilled by the node feature distillation module for inference. Detailed descriptions of the baseline methods and specific experimental settings can be found in Appendix C.

---

[2]https://www.bonanza.com/
[3]https://grouplens.org/datasets/movielens/
[4]https://jmcauley.ucsd.edu/data/amazon/index.html
[5]https://jmcauley.ucsd.edu/data/amazon_v2/index.html

Table 1: Performance (average $\pm$ standard deviation) on Medium-scale Datasets

| Dataset | Metric | Node2vec | SGCN | SGCL | SGA | SBGNN | SBGCL | SELO | SBSN | SBSN-node |
|---|---|---|---|---|---|---|---|---|---|---|
| Bonanza | AUC | 59.3 | 58.7 | 58.4 | 59.6 | 57.7 | 59.0 | 58.5 | $92.0 \pm 1.4$ | **92.5** |
| | Bin-F1 | 98.8 | 89.6 | 98.7 | 93.5 | 96.2 | 97.3 | 98.4 | $98.8 \pm 0.7$ | **99.3** |
| | Mac-F1 | 49.4 | 48.7 | 51.4 | 51.4 | 54.0 | 55.8 | 58.0 | $81.4 \pm 6.2$ | **85.9** |
| | Mic-F1 | 97.7 | 81.4 | 97.4 | 88.0 | 92.7 | 94.7 | 96.9 | $97.7 \pm 1.3$ | **98.6** |
| U.S. House | AUC | 54.4 | 80.8 | 82.4 | 65.1 | 84.8 | 81.0 | 84.6 | $\mathbf{88.0 \pm 0.4}$ | 85.0 |
| | Bin-F1 | 65.9 | 82.7 | 83.5 | 71.4 | 85.6 | 81.1 | 85.8 | $\mathbf{88.8 \pm 0.2}$ | 85.9 |
| | Mac-F1 | 48.7 | 80.8 | 82.4 | 62.1 | 84.7 | 80.7 | 84.6 | $\mathbf{88.0 \pm 0.4}$ | 85.0 |
| | Mic-F1 | 54.5 | 80.9 | 82.4 | 66.1 | 84.7 | 80.7 | 84.7 | $\mathbf{88.0 \pm 0.3}$ | 85.0 |
| Amazon-Book | AUC | 52.6 | 59.3 | 61.3 | 53.6 | 60.3 | 63.7 | 65.2 | $\mathbf{74.5 \pm 0.1}$ | 73.4 |
| | Bin-F1 | 88.8 | 69.3 | 71.0 | **89.1** | 72.0 | 73.4 | 86.5 | $80.9 \pm 0.2$ | 81.0 |
| | Mac-F1 | 44.4 | 50.4 | 50.2 | 52.5 | 55.2 | 58.7 | 65.2 | $\mathbf{66.6 \pm 0.2}$ | 66.2 |
| | Mic-F1 | 79.8 | 58.2 | 60.4 | **80.7** | 61.2 | 64.0 | 78.3 | $72.7 \pm 0.3$ | 72.7 |
| ML-1M | AUC | 51.0 | 63.2 | 63.2 | 56.6 | 65.2 | 68.5 | 73.5 | $\mathbf{74.3 \pm 0.2}$ | 73.7 |
| | Bin-F1 | 73.0 | 65.2 | 67.3 | 73.5 | 69.9 | 70.2 | **77.5** | $77.3 \pm 0.2$ | 76.5 |
| | Mac-F1 | 36.6 | 61.5 | 66.2 | 48.6 | 65.3 | 67.8 | 73.5 | $\mathbf{74.1 \pm 0.2}$ | 73.4 |
| | Mic-F1 | 57.5 | 62.7 | 65.2 | 61.9 | 67.4 | 68.0 | 74.1 | $\mathbf{74.5 \pm 0.2}$ | 73.7 |

## 5.3 Performance on Link Sign Prediction

As shown in Table 1, our proposed model SBSN achieves the best performance in terms of the AUC metric on all datasets, outperforming the state-of-the-art model by 28.5%, 3.2%, 9.3%, and 0.8% on the midium-scale datasets. It can be observed that our model exhibits significant improvements, particularly on imbalanced datasets (Bonanza, Amazon-Book). On large-scale datasets, due to CPU time (see Appendix E) or GPU memory overflow, the performance of SGRL methods and SELO could not be tested. However, our model still achieved strong performance (see Appendix G).

Additionally, SBSN-node directly uses the node features obtained through the node feature distillation module for link sign prediction. The results show that the performance of SBSN-node incurs only minimal loss compared to subgraph-based feature extraction. The detailed analysis of inference efficiency in Appendix D highlights the substantial improvements in inference efficiency afforded by our node distillation module.

## 5.4 Ablation Study and Parameter Study

In the ablation experiments, as shown in Table 2, we tested the module ablation effects on the U.S. House dataset. Firstly, we removed the edge features $H_{u,v}^E$ from the Directed Sign-Flow Passing Module and replaced them with the node features $\mathbf{Z}_v$. The model experienced a significant per-

Table 2: Ablation Study on U.S. House Dataset

| models | AUC | Bin-F1 | Mac-F1 | Mic-F1 |
|---|---|---|---|---|
| SBSN | 88.0 | 88.8 | 88.0 | 88.0 |
| SBSN w/o edge | 87.1 | 87.9 | 87.0 | 87.1 |
| SBSN_GCN | 85.6 | 86.5 | 85.6 | 85.7 |

formance loss due to the loss of edge feature granularity in the information aggregation process. Next, we replace the Directed SignFlow Passing Module with a GCN Mean Aggregator, which also resulted in a certain degree of performance degradation.

Figure 4(a) presents the performance of our model in terms of AUC on the ML-1M dataset with varying values of the subgraph maximum nodes limit. The results indicate a positive correlation, with performance steadily improving as the limit increases. However, the marginal gains in AUC diminish as the limit increases, indicating that while larger neighborhood sizes improve performance, diminishing returns occur beyond a certain point.

Additional ablation studies (see Appendix H) demonstrated that our Topological Subgraph Extractor utilizes information more efficiently than other sampling algorithms, established that different sampling strategies offer dataset-specific advantages, and identified the comprehensive utilization of node features as a primary driver for the superior performance observed on the Bonanza dataset.

## 5.5 Memory Efficiency Analysis

In this section, we compare the GPU memory consumption of our model with SGRL-based methods on subsets of the ML-10M dataset of varying sizes. As shown in Figure 4(b), SBGNN and SBGCL

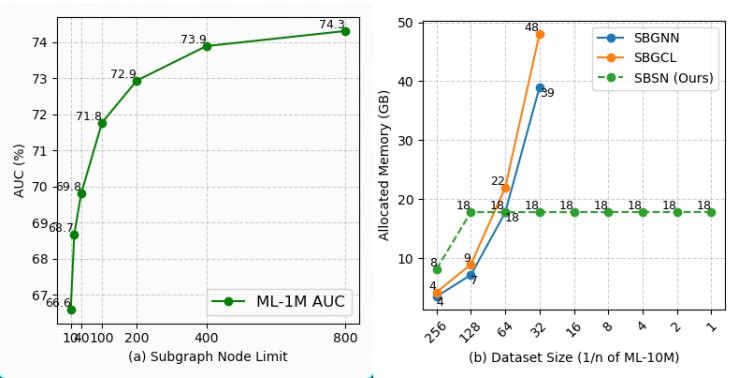

Figure 4: Parameter Experiment and GPU Memory Analysis

exhibit rapidly increasing memory consumption as dataset size grows. This is because SGRL-based methods perform message passing through GNN layers across the entire bipartite graph, resulting in rapid memory growth as the dataset size increases. In contrast, SBSN demonstrates a significantly more stable memory footprint across different dataset sizes. This highlights the memory efficiency of our method, as it processes smaller topological subgraphs rather than the entire graph during message passing. By leveraging this efficient subgraph sampling and message-passing strategy, SBSN achieves a balance between memory efficiency and model performance, enabling its application to large-scale bipartite graphs. The memory analysis presented in Appendix F demonstrates that, for large-scale datasets, the GPU memory consumption of our method scales linearly with the number of nodes in the dataset.

# 6 Conclusion

In this paper, we introduce SBSN to address the challenges in link sign prediction for signed bipartite graphs. Experiments on real-world datasets demonstrate that SBSN outperforms existing baselines in predictive performance while maintaining computational efficiency. Importantly, the scalability of SBSN makes it a versatile solution for large-scale applications.

While our results highlight the effectiveness of subgraph embedding, future work could explore improvements in three key areas. First, developing more efficient message passing modules could further optimize computational complexity. Second, incorporating self-explainability techniques may improve model transparency. Finally, extending SBSN to dynamic signed bipartite graphs would enable real-time prediction and adaptation to evolving network structures.

# 7 Acknowledgment

This work is supported by the projects of Beijing Nova Interdisciplinary Program (20240484647) and National Natural Science Foundation of China (No. 62376012), which is also a research achievement of State Key Laboratory of Multimedia Information Processing and Key Laboratory of Science, Technology and Standard in Press Industry (Key Laboratory of Intelligent Press Media Technology).

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

# A    Complexity Analysis

We provide a detailed analysis of the computational complexity of our proposed SBSN model, focusing on both space and time requirements.

## A.1    GPU memory Complexity

The gpu memory requirements for SBSN consist of storage for model parameters and node features, as well as dynamic memory for intermediate variables during batch processing. The primary storage for node features amounts to $N \cdot d$, where $N$ is the total number of nodes and $d$ is the feature dimension. The architecture of our model, featuring three SignFlow Aggregator modules (each with fixed linear layers and one edge attention layer), contributes an additional $O(d^2)$ to the learnable parameter space.

During batch processing, intermediate variables are generated. For a batch of size $B$ with subgraphs of size $T$, subgraph node features require $B \cdot T \cdot d$ space. The computations of the Edge Attention module, including $H_{u,v}^E$ and pooled attention values, have an upper bound of $O(B \cdot T^2 \cdot d^2)$ in a worst-case complete graph scenario within the subgraph. Other linear layers and attention mechanisms contribute terms on the order of $B \cdot T^2 \cdot d$ and $B \cdot T \cdot d^2$. Consequently, the overall memory complexity for SBSN is $N \cdot d + O(B \cdot T^2 \cdot d^2)$. This contrasts with standard GCN approaches (often used in SGRL methods), which may require $O(E)$ memory per message aggregation step across all $E$ edges, a prohibitive cost for large-scale graphs because $E = O(N^2)$.

The memory consumption varies slightly between phases. During training, the complexity is $N \cdot d + O(B \cdot T^2 \cdot d^2)$. For inference without Node Distillation, it remains similar. However, when Node Distillation is applied for inference, the memory consumption for the active computation part is significantly reduced. Since predictions for a pair $(u, v)$ primarily involve their features and a simpler distilled head, the batch-dependent part related to $T$ diminishes, leading to a memory footprint closer to $N \cdot d + O(B \cdot d^2)$ for the relevant computations. By carefully selecting batch size $B$ and subgraph size $T$, SBSN can efficiently process very large graphs; our estimations suggest it can handle datasets with up to $N = 1.7 \times 10^8$ nodes on a single 80GB GPU under typical experimental settings with $T = 800$. For detailed memory calculation, please refer to Appendix F

## A.2    Inference Time Complexity

We analyze the time complexity on a per-query basis, highlighting the efficiency gains from Node Feature Distillation during inference.

For a single query processed *without Node Distillation*, the computation involves two main stages. First, the sampling stage, which constructs the topological subgraph, has a worst-case time complexity of $O(T^2 \cdot \log(N))$, where the bottleneck is the induced subgraph extraction (assuming, for instance, scanning through neighbor dictionaries in dense regions for $T$ nodes). Second, the message passing stage within the sampled subgraph, analogous to its space requirements, has an upper bound of $O(T^2 \cdot d^2)$. Thus, the total time complexity per query in this mode is $O(T^2(\log(N) + d^2))$.

In contrast, when inference is performed *with Node Distillation*, only the features of the queried nodes $u$ and $v$ are processed through the distilled model head (typically a few MLP layers). This dramatically reduces the time complexity per query to $O(d^2)$. This level of efficiency is comparable to that of many SGRL methods during their inference phase, demonstrating a significant practical advantage of our distillation approach.

# B    Dataset Details

Each dataset contains a different number of nodes and edges, as well as positive and negative relationships between them, with detailed statistics listed in Table 3. During the experiments, we follow the experimental setup in [5], randomly splitting the links of each dataset into three parts: 10% for testing, 5% for validation, and the remaining 85% for training. For the large-scale dataset, we split the data into 8% for testing, 2% for validation, and 90% for training, and average the results over three runs. To ensure the stability and reliability of the experimental results, We run with different train-val-test splits for 5 times to get the average scores.

Table 3: Statistics on Signed Bipartite Networks

| | Bonanza | U.S. House | ML-1M | Amazon-Book | ML-10M | ML-32M | Amazon-Book-51M |
|---|---|---|---|---|---|---|---|
| $|U|$ | 7,919 | 515 | 6,040 | 35,736 | 69,878 | 200,948 | 2,930,451 |
| $|V|$ | 1,973 | 1,281 | 3,952 | 38,121 | 10,677 | 84,432 | 15,362,619 |
| $|E|$ | 36,543 | 114,378 | 1,000,209 | 1,960,674 | 10,000,054 | 32,000,201 | 51,311,613 |
| $\% \ |E^+|$ | 0.980 | 0.540 | 0.575 | 0.806 | 0.589 | 0.821 | 0.848 |
| $\% \ |E^-|$ | 0.020 | 0.460 | 0.425 | 0.194 | 0.411 | 0.179 | 0.152 |

## C  Baselines and Experiment Setting Details

**Node2vec baseline**:  Node2vec [7] learning low-dimensional vector representations of nodes in a network, where these vectors can capture the network structure information and neighborhood characteristics of the nodes. To predict the sign of links, we fuse the embeddings of the connected nodes and utilize a logistic regression model with binary cross-entropy loss.  We experimented with various fusion methods—including sum, L1, L2, and concatenation—and explored different hyperparameters for Node2Vec, specifically setting embedding_dim to {16, 32, 64} and walk_length to {20, 30, 40}. After thorough evaluation, we determined that the optimal combination was the sum fusion method, with embedding_dim set to 64 and walk_length set to 40.

**Signed Graph Neural Networks**: SGCN [24] generalize GCN [14] and GAT [27] to signed graphs based on message-passing mechanisms and balance theory. SGCL [25] is the first research to employ graph contrastive learning on unipartite signed graphs. SBGNN [9] designs a new message-passing mechanism for signed bipartite graphs, which is our most competitive competitor.  SBGCL [32] proposes a contrastive learning method for robust signed graph representation learning. SGA[31] proposed a novel data augmentation approach tailored for SGNNs.

**Subgraph-based Networks**: SELO [6] extracts enclosing subgraphs for each target pair of nodes and encodes these subgraphs into vectors.

For a fair comparison, we set all the node embedding dimensions to 32, which is the same as that in SBGNN [9] and SBGCL [32], for all embedding-based methods. For other parameters in baselines, we follow the recommended settings in their original papers. We run up to 12,800,000 subgraph samples with $K = 4$ and $T = 800$ for SBSN for training and choose the model that performs the best in AUC metrics on the validation set. For our SBSN, we use PyTorch to implement it. We use the Adam optimizer with an initial learning rate of 0.002 and a weight decay of 1e-5. We set the maximum number of nodes in the topological subgraph to 800. Our experiments are conducted on an A100 platform.  Additionally, we derived SBSN-node by utilizing the node feature distillation module.

For SGA, we employ GSGNN[18] as the backbone model, which was reported as the top-performing architecture in its original publication. Regarding the hyperparameters, specifically $\epsilon_{del}^+$, $\epsilon_{del}^-$, $\epsilon_{add}^+$, and $\epsilon_{add}^-$, we searched for optimal values of $\epsilon_{del} \in \{0.1, 0.2, 0.3\}$ and $\epsilon_{add} \in \{0.94, 0.98, 0.99\}$ across different datasets.  Compared to the direct application of the backbone model, The data augmentation of SGA significantly enhanced model performance. Nevertheless, due to the backbone model being a general graph model (i.e., not specifically designed for bipartite structures), SGA exhibited considerable variance in performance on bipartite graph datasets.

During the GPU memory test, we monitored the peak GPU memory usage by tracking the maximum allocated memory with `torch.cuda.memory_allocated`.

The evaluation task is link sign prediction, which is a binary classification problem. We use AUC (Area Under the Curve), Binary-F1, Macro-F1, and Micro-F1 to evaluate the results. These metrics are widely used in existing work [9, 32].

AUC is particularly important as it measures the ability of the model to distinguish between positive and negative classes. It provides a single scalar value that summarizes the performance of the model across all classification thresholds. Thus, we use AUC as the primary evaluation metric for assessing the link sign prediction performance of models.

## D  Inference Efficiency Analysis

Table 4 presents the inference time of different models on the ML-1M dataset. We tested the time consumption of performing inference on the entire ML-1M dataset for each model, where link sign prediction is performed for all edges in the dataset. As seen, SBGCL achieves the fastest inference time of 0.00014 seconds, followed by SBGNN and SBSN-node, both of which require 0.0043 seconds per dataset.

In contrast, SBSN, which relies on subgraph extraction and encoding during inference, requires 2658 seconds per dataset, demonstrating its substantial computational overhead compared to node feature-based methods. SELO, due to its lack of constraints on subgraph size, results in a prohibitive inference time of 354353 seconds per dataset, making it unsuitable for practical applications.

SBSN-node, leveraging the node feature distillation module, significantly improves inference efficiency while retaining comparable performance to SBSN. On the ML-1M dataset, the inference speed is improved by a factor of 618139x compared to SBSN. This highlights the advantage of integrating the distillation module for scenarios requiring rapid inference or node-based downstream tasks.

Table 4: Inference Time (s) of Different Models on the ML-1M Dataset

| Metric | SBGCL | SBGNN | SBSN | SELO | SBSN-node |
|---|---|---|---|---|---|
| Inference Time (s) | 0.00015 | 0.0043 | 2658 | 354353 | 0.0043 |

## E  Time Estimation

On ML-10M, our model required 27 hours for training. In contrast, SELO lacks strict constraints on subgraph size during the matrix computation phase, resulting in large-scale matrix operations. Consequently, the estimated training time for SELO is approximately 152,000 hours, which is impractical and unacceptable.

## F  Analysis of GPU Memory Consumption

Table 5: GPU Memory Consumption (MiB) across Different Models and Datasets

| Model | U.S. House | ML-1M | ML-10M | Amazon-Book | Amazon-Book-51M |
|---|---|---|---|---|---|
| SBSN | 5404 | 18273 | 18297 | 18282 | 24959 |
| SBGNN | 951 | 22823 | OOM | OOM | OOM |
| SBGCL | 1178 | 28031 | OOM | OOM | OOM |

As shown in Table 5, we evaluate the GPU memory consumption of three models (SBSN, SBGNN, and SBGCL) across four datasets: U.S. House, ML-1M, ML-10M, Amazon-Book, and Amazon-Book-51M. All experiments were performed using an NVIDIA A100 80GB PCIe GPU, ensuring a consistent computational environment.

SBSN exhibits superior scalability and efficiency compared to other models. It successfully operates on all four datasets with memory usage ranging from 5,404 MiB (U.S. House) to 24,959 MiB (Amazon-Book-51M), demonstrating its capability to handle large-scale datasets while keeping memory consumption manageable. Notably, compared to the 18,273 MiB used on ML-1M, the memory usage on Amazon-Book-51M increases by 6,686 MiB. This increment precisely corresponds to the additional memory required to store node feature tensors. Specifically, the theoretical memory cost can be computed as

$$(|U| + |V|) \times d \times C_{\text{fp32}} \times 3 = 18{,}293{,}070 \times 32 \times 4 \times 3 \text{ bytes} \approx 6{,}699 \text{ MiB},$$

where $d$ denotes the feature dimension, $C_{\text{fp32}}$ the number of bytes per 32-bit float, and the factor 3 accounts for the parameter, gradient, and optimizer states.

In contrast, SBGNN and SBGCL encounter out-of-memory (OOM) issues on the larger datasets (ML-10M and Amazon-Book), highlighting their limitations in scalability. Although SBGNN uses 951 MiB on the smallest dataset (U.S. House), its memory consumption rises significantly to 22823

MiB on ML-1M, suggesting poor memory efficiency as dataset size increases. Similarly, SBGCL uses 1178 MiB on U.S. House but fails to execute on datasets larger than Amazon-Book.

Overall, the results highlight SBSN's robust memory management and scalability, enabling it to effectively handle large-scale datasets where competing models fail due to excessive GPU memory demands.

## G   Performance on Large Scale Datasets

Table 6: Performance (average $\pm$ standard deviation) on Large-scale Datasets

| Dataset | Metric | Node2vec | SGCN | SGCL | SGA | SBGNN | SBGCL | SELO | SBSN | SBSN-node |
|---|---|---|---|---|---|---|---|---|---|---|
| ML-10M | AUC | 54.9 | | | OOM | | | TLE | **75.2 ± 0.1** | 74.9 |
| | Bin-F1 | 65.6 | | | OOM | | | TLE | **79.4 ± 0.2** | 78.0 |
| | Mac-F1 | 52.8 | | | OOM | | | TLE | **75.2 ± 0.2** | 74.5 |
| | Mic-F1 | 56.3 | | | OOM | | | TLE | **75.9 ± 0.2** | 75.0 |
| ML-32M | AUC | 56.3 | | | OOM | | | TLE | **78.3 ± 0.1** | 77.5 |
| | Bin-F1 | **90.1** | | | OOM | | | TLE | 86.0 ± 0.1 | 85.3 |
| | Mac-F1 | 45.5 | | | OOM | | | TLE | **71.3 ± 0.1** | 70.4 |
| | Mic-F1 | **82.0** | | | OOM | | | TLE | 78.8 ± 0.1 | 77.9 |
| Amazon-Book-51M | AUC | 57.9 | | | OOM | | | TLE | **73.1 ± 0.1** | 72.5 |
| | Bin-F1 | **91.8** | | | OOM | | | TLE | 82.8 ± 0.2 | 82.6 |
| | Mac-F1 | 45.9 | | | OOM | | | TLE | **64.1 ± 0.2** | 63.9 |
| | Mic-F1 | **84.8** | | | OOM | | | TLE | 73.9 ± 0.2 | 73.9 |

## H   Additional Ablation Study

Table 7: Performance (AUC) of Different SBSN Variants

| Variant | Bonanza | U.S. House | Amazon-book | ML-1M | Avg. AUC |
|---|---|---|---|---|---|
| Base | 92.0 | 88.0 | 74.5 | 74.3 | 82.2 |
| T=1400, w/o connect | 91.0 | 83.5 | 71.5 | 72.5 | 79.3 |
| w/o node | 69.6 | 87.5 | 74.0 | 73.4 | 76.3 |
| T=100, low | 91.1 | 82.7 | 73.1 | 72.4 | 79.3 |
| T=100, rand | 92.0 | 84.7 | 73.5 | 71.7 | 80.2 |
| T=100, high | 93.0 | 84.0 | 73.2 | 70.9 | 80.0 |

Below, we provide a supplementary experimental table that evaluates different variants of SBSN. The experimental settings remain consistent with those described in the main text.

**w/o connect:** Replacing the Topological Subgraph Extractor with independent random 2-hop neighbor sampling (resulting in $T = 1400$) produces performance similar to SGRL. This comparison illustrates that our sampling strategy, which focuses on extracting connected topological subgraphs (with $T = 800$), is critical to achieving higher performance.

Traditional subgraph extraction algorithms[1, 6] sample a subgraph $G_{\text{sub}}$ from several layers of the neighborhood around the target nodes $u$ and $v$. These methods typically perform undirected message passing over the extracted subgraph. However, these subgraphs include numerous nodes that do not appear in any simple path connecting $u$ and $v$ within $G_{\text{sub}}$, leading to unnecessary computational overhead. In contrast, our method extracts topological subgraphs and performs directed message passing, effectively avoiding this issue. The subgraph embedding aggregates information only from all simple paths in $G_{\text{sub}}$ that connect $u$ and $v$.

For example, on the ML-1M dataset—the densest graph in our experiments—we found that a two-hop neighbor sampling with $T = 1400$ produces results similar to those from our Topological Subgraph Extractor with $T = 150$. This comparison highlights the high efficiency of our sampling algorithm.

**w/o node:** Removing node features and replacing them with random vectors leads to a performance drop, especially on the Bonanza dataset. This suggests that the high-quality global node features learned in the local subgraph training contribute significantly to the performance of SBSN. Our subgraph sampling method guarantees that every node in the sampled subgraph lies on a simple path connecting the target nodes $u$ and $v$. In contrast, simple two-hop neighbor sampling for $u$ and $v$

separately does not offer such a guarantee, which significantly reduces the efficiency in information utilization.

**Different Sampling Strategies:** The experiments with fixed subgraph size ($T = 100$) and varying heuristic sampling strategies (low, random, high degree) indicate that the optimal strategy can differ by dataset. We chose the random sampling strategy for the baseline due to its overall superior performance across datasets.

# I Theoretical Calculation for the Number of Topological Subgraph Layers

Let $T$ be the total number of nodes in the topological subgraph and $p$ be the probability of an edge existing between any two distinct nodes (i.e., the graph density). Because the first layer only contains a single node $u$ and the last layer only contains node $v$, we let $K^* = K - 2$ as the number of mid-layers. The $T$ nodes are divided equally into $K^*$ mid-layers, so each mid-layer contains $n_k = T/K^*$ nodes.

To calculate the expected number of such paths, denoted by $E(K^*)$, we first consider the number of possible unique sequences of nodes, one from each mid-layer. There are $(T/K^*)$ choices for a node from each mid-layer. Thus, there are $(T/K^*)^{K^*}$ distinct potential sequences of $K^*$ nodes, one from each layer.

For any single such sequence $(v_1, v_2, \ldots, v_{K^*})$ to constitute an actual path, there must be $K - 1$ edges present: $(v_1, v_2), (v_2, v_3), \ldots, (v_{K^*-1}, v_{K^*})$. Assume the sampled subgraph as an Erdős-Rényi random graph $G(T, p)$, the probability of any specific edge existing between nodes in different layers is $p$, and these edge existences are independent. Therefore, the probability that all $K^* - 1$ required edges for a specific sequence exist is $p^{K^*-1}$.

By the linearity of expectation, the total expected number of paths $E(K^*)$ is the product of the number of potential node sequences and the probability of any one such sequence forming a path:

$$E(K) = \left(\frac{T}{K^*}\right)^{K^*} p^{K^*-1} \tag{3}$$

To find the optimal $K^*$ that maximizes $E(K^*)$, it is often easier to maximize its natural logarithm, $\ln(E(K^*))$, since the logarithm is a monotonically increasing function.

$$\ln(E(K^*)) = \ln\left(\left(\frac{T}{K^*}\right)^{K^*} p^{K^*-1}\right)$$

$$= K^* \ln\left(\frac{T}{K^*}\right) + (K^* - 1)\ln(p)$$

$$= K^*(\ln(T) - \ln(K^*)) + (K^* - 1)\ln(p)$$

Treating $K^*$ as a continuous variable for the purpose of optimization, we differentiate $\ln(E(K^*))$ with respect to $K^*$ and set the derivative to zero:

$$\frac{d}{dK^*}\ln(E(K^*)) = \frac{d}{dK^*}[K^* \ln(T) - K^* \ln(K^*) + K^* \ln(p) - \ln(p)]$$

$$= \ln(T) - (\ln(K^*) + K^* \cdot \frac{1}{K^*}) + \ln(p) - 0$$

$$= \ln(T) - \ln(K^*) - 1 + \ln(p)$$

Setting the derivative to zero to find the critical points:

$$\ln(T) - \ln(K^*) - 1 + \ln(p) = 0$$
$$\ln(K^*) = \ln(T) + \ln(p) - 1$$
$$\ln(K^*) = \ln(Tp) - \ln(e)$$
$$\ln(K^*) = \ln\left(\frac{Tp}{e}\right)$$

Thus, the value of $K^*$ that maximizes $E(K^*)$ when $K^*$ is treated as a continuous variable is:

$$K^*_{\text{conti}} = \frac{Tp}{e} \tag{4}$$

where $e$ is the base of the natural logarithm.

In practice, $K^*$ must be an integer, and typically $K^* \geq 2$ (for a path to exist between at least two layers).

In our experiments on real-world datasets, we empirically investigated the optimal $K = K^* + 2 = Max\{2, \lfloor \frac{Tp}{e} \rfloor\} + 2$, with the additional constraint that $K$ is an even number. Under a default setting for the sampled subgraph size $T = 800$, we calculated that the optimal $K$ value was 4 for a majority of the datasets, including Bonanza, Amazon-book, ML-10M, ML-32M, and Amazon-book-51M. Consequently, we uniformly adopted $K = 4$ as a practical setting across our main experiments for consistency and based on these findings. Furthermore, for dense datasets where theoretical calculations or initial observations suggested an optimal $K > 4$ (U.S. House, ML-1M), we also experimented with $K = 6$. However, this increase in the number of layers did not yield an improvement in final performance, which we attribute to the over-smoothing phenomenon[22] that can affect Graph Neural Networks with deeper multi-hop message passing.

Table 8: Statistics on Signed Bipartite Networks

|  | Bonanza | U.S. House | ML-1M | Amazon-Book | ML-10M | ML-32M | Amazon-Book-51M |
|---|---|---|---|---|---|---|---|
| $|U|$ | 7,919 | 515 | 6,040 | 35,736 | 69,878 | 200,948 | 2,930,451 |
| $|V|$ | 1,973 | 1,281 | 3,952 | 38,121 | 10,677 | 84,432 | 15,362,619 |
| $|E|$ | 36,543 | 114,378 | 1,000,209 | 1,960,674 | 10,000,054 | 32,000,201 | 51,311,613 |
| $p$ | 0.002339 | 0.173375 | 0.041902 | 0.001439 | 0.013403 | 0.001886 | 0.000001 |
| $T$ | 800 | 800 | 800 | 800 | 800 | 800 | 800 |
| $K^*_{\text{conti}}$ | 0.69 | 51.02 | 12.33 | 0.42 | 3.94 | 0.56 | 0.00 |
| $K$ | 4 | 53 | 14 | 4 | 4 | 4 | 4 |

