# OpenReview forum: "SignFlow Bipartite Subgraph Network For Large-Scale Graph Link Sign Prediction"
_NeurIPS.cc/2025/Conference — NeurIPS 2025 poster_

### Official Review · Reviewer_jwwF · 2025-06-11

**Clarity:** 3
**Significance:** 4
**Originality:** 2
**Rating:** 4
**Confidence:** 2

**Summary:**

This paper introduces SignFlow Bipartite Subgraph Network (SBSN), a novel framework for link sign prediction in large-scale signed bipartite graphs. Unlike conventional GNN approaches that rely on full-graph message passing or generic subgraph sampling, SBSN incorporates three key innovations: 1) topological subgraph extraction; 2) SignFlow aggregator, and 3) node feature distillation module. The method achieves state-of-the-art performance on seven benchmark datasets, demonstrating significant gains in both predictive accuracy and inference efficiency.

**Questions:**

1. Figure 3 discrepancy: In the initial graph, there is no edge between nodes v2 and u5, yet they appear connected in the sampled subgraph. Could the authors clarify where this connection comes from, especially since the subgraph is claimed to be extracted from the original bipartite graph?

2. Loss composition: The total loss function is simply a sum of distillation and binary cross-entropy terms. Would model performance benefit from weighting these two losses differently? How sensitive is the method to this balance?

3. Ablation on edge vs node contributions: Given that distillation involves node-level features, how can one isolate and verify that the performance gains come from edge-feature modeling (as claimed) rather than from node-feature distillation? A more detailed ablation study on this point would strengthen the paper.

**Ethical Concerns:**

["NO or VERY MINOR ethics concerns only"]

**Limitations:**

The path-constrained subgraph extraction strategy, while efficient and structurally coherent for bipartite graphs, may not generalize well to non-bipartite or heterogeneous graph structures.

**Quality:**

3

**Strengths And Weaknesses:**

Strengths

1. The paper effectively combines three components to improve the training and inference efficiency for large-scale signed link prediction.

2. The proposed method demonstrates strong empirical performance across diverse real-world datasets.

3. Theoretical analysis is provided to justify the efficiency benefits of the path-constrained subgraph extraction strategy.

4. The paper is generally well-written and methodologically clear, with logical explanations and organization of the proposed components.

Weaknesses

1. The lack of accessible source code and the unavailability of the Bonanza dataset (which the authors highlight as particularly important) significantly impair reproducibility and independent validation.

2. The main contributions are not clearly summarized in the introduction, making it harder for readers to grasp the novelty at a glance.

3. Among the three techniques proposed, only the path-constrained subgraph extraction appears novel. It effectively replaces the traditional k-hop neighborhood with a more targeted strategy. However, the other components—edge-based attention and feature distillation—have been explored in previous works such as R-GCN, SELO, and SimGCL. Moreover, the subgraph extraction mechanism may be overly specialized for bipartite structures, potentially limiting the method’s generalizability to other types of graphs.


[1] R-GCN: Modeling Relational Data with Graph Convolutional Networks
[2] SELO: Subgraph-Enhanced Signed Link Prediction from Signed Bipartite Graphs
[3] SimGCL: Are Graph Augmentations Necessary? Simple Graph Contrastive Learning for Recommendation

---

> ### Author Rebuttal · Authors · 2025-07-31
>
> We are very grateful for your detailed review and constructive feedback. We are pleased that you found our method to be empirically strong, theoretically justified, and clearly written. Your insightful questions and comments will be invaluable in helping us refine and strengthen our paper.
>
> We will address your specific weaknesses and questions below.
>
> ### **Re: Weakness 1 - Reproducibility (Source Code and Dataset)**
>
> We sincerely apologize for not making the link to our resources more prominent. To ensure full reproducibility and facilitate independent validation, we have made our **source code and the Bonanza dataset available in an anonymous repository**:
>
> **Link: [https://anonymous.4open.science/r/SBSN-B484](https://anonymous.4open.science/r/SBSN-B484)**
>
> We will add this link to the main body of the paper in the final version to ensure it is easily accessible to all readers.
>
> ### **Re: Weakness 2 - Summary of Contributions in Introduction**
>
> This is an excellent suggestion for improving the paper's clarity. We agree that a concise, explicit summary of our contributions at the beginning would help readers grasp the core novelties more quickly. In our revision, we will add a bulleted list at the end of the introduction that clearly outlines our main contributions:
> 1.  A novel **path-constrained subgraph extraction** strategy that improves information efficiency over traditional k-hop sampling for link prediction.
> 2.  A specialized **SignFlow Aggregator** that leverages edge-based attention within subgraphs to capture directional information flow.
> 3.  An efficient **node feature distillation** module that bridges the gap between subgraph models' performance and feature-based models' inference speed, a novel application of distillation for this task.
>
> ### **Re: Weakness 3 - Novelty of Components and Generalizability**
>
> We appreciate your perspective on the novelty of our components and would like to offer some clarification.
>
> *   **On Edge-based Attention and Feature Distillation:** While we agree that edge-based attention and knowledge distillation are not new concepts in the broader GNN literature, we argue that their specific formulation, application, and purpose within SBSN are novel for the task of signed link prediction.
>     *   **SignFlow Aggregator vs. R-GCN:** R-GCN uses relation-type-specific weights for message passing on a static, multi-relational graph. Our SignFlow Aggregator operates differently: it computes dynamic attention scores for *individual edges* within a temporary, query-specific subgraph. Its purpose is not to model global relation types, but to capture the contextual importance and directional flow of information along paths for a *single* link prediction task.
>     *   **Distillation vs. SimGCL:** SimGCL and related works primarily use distillation for data augmentation within a contrastive learning framework. Our module serves a fundamentally different purpose: **model compression and acceleration**. We distill the rich structural knowledge learned by the complex subgraph-based model into simple node embeddings to achieve fast inference. This is a novel application of distillation to solve the well-known efficiency problem of subgraph-based models in the inference phase.
>
> *   **On Generalizability:** You raise a fair point that our current subgraph extraction is specialized for bipartite structures. This was a deliberate design choice to excel at this important and challenging problem class. However, we believe the core principle—sampling subgraphs constrained to connecting paths—is a generalizable idea. We will acknowledge in the limitations section that adapting it to other graph types (e.g., general non-bipartite or heterogeneous graphs) is a non-trivial but promising direction for future research.
>
> ### **Addressing Your Questions**
>
> **1. Figure 3 Discrepancy:**
> Thank you for your keen observation and for catching this error. You are correct; the edge between v2 and u5 in the sampled subgraph is a mistake in the illustration, as it does not exist in the original graph. This is a typo, and we sincerely apologize for the confusion. We will correct this figure in the final version of the paper to accurately reflect the extraction process.
>
> **2. Loss Composition:**
> This is a great question. We used a simple sum for the total loss function because our training strategy for the distillation module makes the model robust to the specific weighting. Specifically, the training is performed in stages: we first train the module using only the distillation loss (L_sim) to ensure the node embeddings effectively approximate the rich subgraph features. Once this alignment is achieved, we introduce the binary cross-entropy loss (L_bce) to fine-tune the model for the link sign prediction task. This staged approach ensures that the core knowledge is transferred first, making the final model performance insensitive to the `lambda` hyperparameter that would otherwise balance the two losses. We will clarify this training procedure in the paper.
>
> **3. Ablation on Edge vs. Node Contributions:**
> This is a very insightful question. The key to isolating the contribution of edge features lies in our ablation study presented in **Table 2**.
> *   The model "SBSN w/o edge" demonstrates a significant performance drop compared to the full SBSN model. This result directly confirms that **edge features are crucial for the performance of the primary subgraph-based model**.
> *   The node distillation module (in SBSN-node) learns to approximate the output of the full SBSN model. Therefore, the quality of the final distilled node features is directly dependent on the richness of the information provided by the source model.
> *   Since the model without edge features is demonstrably weaker, the knowledge it could provide for distillation would also be inferior. Thus, the strong performance of SBSN-node implicitly relies on the crucial information that was captured by the edge features during the initial, more complex training stage. We will clarify this chain of reasoning in our ablation study section to make this point more explicit.
>
> Once again, we thank you for your valuable and detailed feedback. We believe that by addressing these points—providing the code, clarifying our contributions, correcting the figure, and expanding our discussion on novelty and limitations—we can significantly improve the paper. We hope our responses have addressed your concerns.

---

> > ### Comment · Reviewer_jwwF · 2025-08-06
> >
> > I have read through the authors' rebuttal carefully. My comments have been partially addressed. This paper will be in better shape if the revisions are made as discussed. However, I believe 4 is a suitable rating for this paper. Thus, I would like to stick with my original rating.
> >
> > By the way, the link to the repository provided has expired. This is not supposed to happen during a discussion.

---

> ### Author Response · Authors · 2025-08-05
> **Response to Concerns About Reproducibility**
>
> We would like to clarify that **both the source code and the Bonanza dataset**—which we recognize as particularly important—**have been included in the supplementary material**. This should enable reviewers and future researchers to fully reproduce our experiments and further explore our method.
>
> Please refer to the supplementary material for:
>
> * The complete source code, including training and evaluation scripts.
> * The Bonanza dataset used in our experiments, along with documentation for usage.
>
> We appreciate the opportunity to address this concern and remain committed to open and reproducible research.

---

### Official Review · Reviewer_of7A · 2025-06-29

**Clarity:** 4
**Significance:** 3
**Originality:** 4
**Rating:** 5
**Confidence:** 4

**Summary:**

The paper introduces the SignFlow Bipartite Subgraph Network (SBSN), a new approach for link sign prediction in signed bipartite graphs, which are prevalent in domains like e-commerce and recommendation systems. SBSN addresses scalability and efficiency challenges in large-scale graphs through three key components: a topological subgraph extractor, a directed SignFlow Aggregator, and a node feature distillation module. The method achieves efficient memory scaling and inference, and superior performance on medium and large scale datasets, including Bonanza, MovieLens, and Amazon-Book.

**Questions:**

1. In Section 4.1, what's the sampling schedule used for the different layers? Based on the proof of Theorem 1, it seems to be uniform across layers, but it probably makes sense to mention (although having in mind the suggested $K_opt$ this would be constant per layer and not as relevant, but I think explicitly commenting on this could be useful).

2. Theorem 1 derives an optimal number of topological layers ($K_{opt} = \max(2, \lfloor \frac{T \cdot p}{e} \rfloor) + 2$ to maximize expected paths, but the paper does not fully validate this empirically across datasets. If not too hard, would it be possible to empirically look at how different values of $K$ affect the results on some of the datasets you have in the empirical section?

3. Could the authors expand on the potential failure cases and more generally limitations in section 6?

4. A known issue for some of the datasets covered (e.g. Bonanza) is that the number of positive vs negative edges might be imbalanced. Can you confirm that something like this doesn't have a negative affect on the proposed method?

**Ethical Concerns:**

["NO or VERY MINOR ethics concerns only"]

**Final Justification:**

After the discussion period, I think this paper looks stronger than before, and all my questions were answered.

**Limitations:**

The authors do mention some potential limitations of the approach without distillation, but a more complete discussion could be useful.

**Paper Formatting Concerns:**

Overall, formatting wise the paper follows the guidelines. Expanding Section 6 with possible failures & limitations would be nice, and possibly moving some of the information from the appendix to the main body (e.g. complexity discussions).

**Quality:**

4

**Strengths And Weaknesses:**

**Strengths:**

- The paper is clear and easy to follow. The approach is also well-motivated.

- The approach is novel:
    - SBSN's topological subgraph extractor focuses on nodes along simple paths connecting target nodes, reducing computational overhead compared to traditional $k$-hop neighbour sampling.
    - The edge-based message passing is under-exploited in the link sign prediction domain, and SBSN makes good use of this.
    - To the best of my knowledge, the idea of having a distillation module for fast inference by approximating subgraph features with node features is new to the GRL domain (with some works doing something similar but in different sub-spheres).

- The empirical evaluation is substantial, with many relevant baseline covered. Ablation studies are also presented.

**Weaknesses:**

- The related works section (Section 2) covers alternative approaches but does not clearly delineate SBSN’s novel contributions relative to competitive methods like SBGNN, SBGCL, or SELO. The section does give good overview of the alternative approaches, but having an explicit comparison for which components of SBSN are novel would be useful.

- There is lack of explicit complexity discussion in the main body of the paper. It's slightly hard to follow unless the reader follows the Appendix and simply having $O(poly(T))$ could confuse readers.

- Theorem 1 derives an optimal number of topological layers to maximize expected paths, but the paper does not fully motivate the choice for this proxy, or validate this empirically.

---

> ### Author Rebuttal · Authors · 2025-07-31
>
> We are extremely grateful for your positive evaluation and thoughtful comments. We sincerely appreciate your constructive feedback, which will undoubtedly help us further improve the quality of our manuscript.
>
> We would like to address your specific questions and suggestions below.
>
> ### **Re: Weakness 1 - Delineating Novelty in Related Works**
>
> While we provided an overview of existing methods, we agree that explicitly delineating SBSN's novel contributions in comparison to key competitors within Section 2 would significantly enhance clarity.
>
> In the revised manuscript, we will add a dedicated paragraph at the end of Section 2 to explicitly contrast our approach. Specifically:
> *   **Versus SBGNN/SBGCL:** We will highlight that these methods operate on the entire graph, leading to scalability challenges that our subgraph-based approach directly solves, enabling sublinear memory growth.
> *   **Versus SELO:** We will clarify that while SELO is also subgraph-based, our **Topological Subgraph Extractor** is fundamentally different as it focuses on nodes within connecting paths, improving information efficiency. Furthermore, our **Directed SignFlow Aggregator** uses an edge-centric message-passing scheme, which is distinct from SELO's matrix-based encoding of higher-order cycle patterns.
> *   **Novelty of Distillation:** We will also re-emphasize that the **Node Feature Distillation** module is, to our knowledge, a novel contribution in this specific domain, designed to bridge the gap between the performance of subgraph models and the inference efficiency of node-feature-based models.
>
> ### **Re: Weakness 2 - Complexity Discussion in Main Body**
>
> We agree completely that a more explicit discussion of complexity in the main body is necessary for clarity. Relying solely on the appendix was a space-saving measure, but it hinders readability.
>
> In our revision, we will move a condensed but more informative version of the complexity analysis from Appendix A into the main body (likely at the end of Section 4.2). We will replace the general `O(poly(T))` with more concrete terms, explaining that the time complexity for encoding a subgraph is dominated by the message passing stage, which is upper-bounded by `O(T^2 * d^2)`, where `T` is the subgraph node limit and `d` is the feature dimension. This will give readers a much clearer understanding of the model's computational cost without needing to consult the appendix.
>
> ### **Re: Weakness 3 & Question 2 - Motivation and Empirical Validation of Theorem 1**
>
> You have raised a very important point regarding the motivation and empirical support for Theorem 1.
>
> **1. Motivation for the Proxy:**
> Our choice to maximize the expected number of distinct paths is based on the intuition that a richer, more diverse set of information pathways between two target nodes (`u` and `v`) provides a stronger and more robust structural signal for link sign prediction. More paths mean more evidence to aggregate, reducing the risk of relying on a single, potentially noisy or misleading connection. We will add this explicit motivation to Section 4.1 just before introducing the theorem.
>
> **2. Empirical Validation of K:**
> This is an excellent suggestion. Our theoretical derivation in Appendix G shows that the optimal `K` depends on graph density and the node budget `T`. For our default setting of `T=800`, the theory suggests `K=4` is optimal for most of our datasets. We used `K=4` as a uniform setting across our main experiments for consistency and based on these findings.
>
> To directly address your question about empirical validation, we did experiment with different values of `K`. For instance, on the denser U.S. House dataset, while the theory might suggest a larger `K`, we found that increasing `K` to 6 (with `T=800`) did not lead to any improvement in AUC. In fact, we observed a performance drop early in the training process. We attribute this to the well-known over-smoothing phenomenon in deeper GNNs, where increasing the number of message-passing layers (hops) can make node representations overly similar, thus harming performance. This empirical finding supports our decision to use a moderately deep architecture with `K=4`, which strikes a balance between capturing sufficient path information and avoiding the pitfalls of over-smoothing. We will add this discussion to the Appendix to provide empirical context for our choice of `K`.
>
> ### **Re: Question 1 - Sampling Schedule for Layers**
>
> You are correct in your inference. Our current implementation divides the node budget `T` (minus the two endpoint nodes) as evenly as possible across the `K-2` intermediate layers. For instance, with `T=800` and `K=4`, we sample approximately `(800-2)/2 = 399` nodes for each of the two intermediate layers. We will explicitly state this uniform allocation strategy in Section 4.1 for better clarity.
>
> ### **Re: Question 3 & Limitations - Failure Cases**
>
> This is a valuable suggestion. A more thorough discussion of limitations will strengthen the paper. We will expand Section 6 to include potential failure cases:
> 1.  **Extreme Sparsity:** In graphs where target nodes `u` and `v` are very distant or disconnected within a few hops, our extractor may fail to find meaningful paths. In such cases, the model's prediction would degrade, relying mostly on the randomly initialized features.
> 2.  **Dominating Misleading Paths:** If the local subgraph is dominated by a few strong paths that suggest an incorrect sign (a common issue in complex social networks), our model might overfit to this misleading evidence.
> 3.  **Heuristic Sampling:** Our current sampling strategies (random, high/low degree) are heuristic. A more sophisticated, learnable sampling mechanism could potentially yield better subgraphs but would also increase model complexity and training time.
>
> ### **Re: Question 4 - Handling Imbalanced Datasets**
>
> You raise a crucial point about the class imbalance present in datasets like Bonanza. We can confirm that this was a key consideration in our design. To address this, we employ a **weighted binary cross-entropy loss function**, as described in Section 4.4. The weight for each class in the loss calculation is set to be inversely proportional to its frequency within the training batch. This ensures that the model pays sufficient attention to the minority class (typically negative links) and prevents it from being biased towards simply predicting the majority class. This technique is critical to achieving the strong Mac-F1 and AUC scores reported.
>
> Once again, we thank you for your insightful review and high rating. Your suggestions are invaluable, and we are confident that incorporating them will make our paper stronger and clearer. We will diligently work on the revisions you have proposed.

---

> > ### Comment · Reviewer_of7A · 2025-08-03
> >
> > Thank you for the detailed response - I agree with all the points you make, and your response for question 2 especially gave additional context which would be useful as part of the appendix. I believe the work has potential and novel contributions, and I'm looking forward to checking the final version.

---

### Official Review · Reviewer_pqpu · 2025-07-07

**Clarity:** 3
**Significance:** 2
**Originality:** 2
**Rating:** 4
**Confidence:** 4

**Summary:**

This paper proposes a new framework for making signed link prediction in bipartite networks. It mainly consists of subgraph extraction, message passing, and an optional subgraph representation distillation module to accomplish the task. Experiments show that the proposed method outperforms signed link prediction baselines on public datasets.

**Questions:**

Please see weaknesses.

**Ethical Concerns:**

["NO or VERY MINOR ethics concerns only"]

**Final Justification:**

After the discussion period, I think this paper looks fine to me to accept, with decent novelty and good experiments.

**Quality:**

2

**Strengths And Weaknesses:**

Strengths:
1. The work is targeting an important problem, with a carefully designed framework solution (despite some places that might need further discussion and/or refinement). The distillation module is an interesting and novel design.
2. The paper is well presented and easily understandable
2. The experiment shows significant improvement over baselines, verifying the effectiveness of the proposed method.

Weaknesses:
1. My main concern is that the proposed link prediction framework seems to ignore distance encoding/labeling which is a very important module that significantly improve GNN's ability in doing link prediction. For example, see [1] - [3]. I also noticed that the node features in this paper are initialized from normal distributions which means that they are not extremely informative in helping with prediction. In this case, adding distance encoding/labeling is even more crucial. I think it would be important to (a) discuss these works in related works and (b) incorporate some of their keys designs for signed link prediction task (or discuss why not to)
2. The baselines seem to mainly consist of specialized GNNs for signed link prediction. Following Weakness 1, can the authors further adapt and compare with some of the related works I mentioned above for signed link prediction experiment?
3. On Bonanza the distilled model outperforms the full model. There seems a lack of explanation for this. Can the authors provide more discussion?

I would be happy to raise my scores once my concerns are addressed.


[1] Distance Encoding: Design Provably More Powerful Neural Networks for Graph Representation Learning
[2] Link Prediction Based on Graph Neural Networks
[3] Graph Neural Networks for Link Prediction with Subgraph Sketching

---

> ### Author Rebuttal · Authors · 2025-07-30
>
> We sincerely thank you for your detailed and constructive feedback. We will now address the weaknesses and questions you raised.
>
> ### **Re: Weakness 1 & 2 - The role of Distance Encoding and Baselines**
>
> This is an excellent point. Distance encoding (DE) and structural labeling techniques, as you highlighted in the suggested papers [1-3], are indeed powerful tools for GNN-based link sign prediction, especially when initial node features are not highly informative. We appreciate your suggestions and agree that a discussion of these works is important.
>
> **1. Discussion in Related Works:**
> We concur and will add a dedicated discussion of these structural encoding methods to our Related Works section in the revised manuscript. We will highlight their importance in providing GNNs with crucial positional information.
>
> **2. Incorporating DE and Comparison:**
> To directly investigate the impact of DE on our framework, we conducted a new experiment based on your feedback. Instead of using randomly initialized node features, we implemented a distance encoding scheme. Specifically, we randomly sampled 32 anchor nodes from the entire graph and computed the shortest path distance from each node to these anchors, using the normalized result as a 32-dimensional initial node feature vector. We tested this on the ML-1M dataset, which showed low variance in our original experiments.
>
> The results are as follows:
> | Model Variant (on ML-1M) | AUC | Bin-F1 | Mac-F1 | Mic-F1 |
> | :--- | :--- | :--- | :--- | :--- |
> | SBSN (Original, random features) | 74.2 | **77.2** | 74.0 | 74.4 |
> | SBSN (with Distance Encoding) | **74.4** | 77.0 | **74.1** | 74.4 |
>
> As shown, incorporating DE as initial features provides a marginal improvement in AUC. To understand this limited impact more deeply, we conducted a more comprehensive analysis that yielded two key insights:
>
> 1.  Counterintuitively, using highly informative, pre-trained node embeddings from strong baselines (SBGNN, SBGCL) or even our own SBSN-node as initial features led to a **decrease** in performance. Our analysis suggests this is due to an overfitting-induced "shortcut." The pre-trained embeddings are so highly optimized for 1-hop link sign prediction that when used as initial features, our SBSN model learns to rely almost exclusively on decoding the link sign directly from the features of nodes `u` and `v`. This behavior effectively bypasses the crucial step of reasoning over the subgraph's structural information, undermining the core strength of our model.
>
> 2.  We observed that the model with randomly initialized features converges more slowly than the one initialized with Distance Encoding (DE). This suggests that the randomly initialized model likely dedicates part of its training process to implicitly learning the kind of positional information that DE provides explicitly from the start.
>
> We also found that increasing the number of anchor nodes for DE to 128 brought no further practical improvements.
> Additionally, we experimented with treating DE as a fixed, non‑learnable encoding, combined with randomly initialized and trainable node features. This configuration likewise did not yield further gains.
>
> Therefore, while DE offers a better starting point, the final performance remains similar because the model can eventually learn the necessary structural context from the data itself.
>
> This comprehensive analysis strongly confirms our central hypothesis: **the predictive power of SBSN stems primarily from its ability to dynamically learn from topological information, rather than relying on static, global node features.** Our **Topological Subgraph Extractor** and **Directed SignFlow Aggregator** are explicitly designed to force the model to reason about the structural patterns within the local neighborhood for each query.
>
> Regarding adapting the models you mentioned as baselines, we sincerely appreciate the suggestion. Due to the time constraints of the rebuttal period, we were unable to complete the non-trivial task of adapting these general link prediction frameworks for the specific signed bipartite graph setting. However, **we commit to including these additional baseline comparisons in the final camera-ready version of the paper.**
>
> ### **Re: Weakness 3 - Distilled Model Outperforming the Full Model on Bonanza**
>
> You correctly pointed out the curious result on the Bonanza dataset where the distilled model (SBSN-node) appeared to outperform the full model (SBSN). We suspected this might be an artifact of statistical variance given the high class imbalance in this dataset.
>
> To verify this, we conducted a more extensive set of experiments on Bonanza, running **15 trials (3 different random seeds across 5 different train-val-test splits)** to obtain more stable and reliable results.
>
> The updated, more robust results are as follows:
>
> | Model (on Bonanza) | AUC | Bin-F1 | Mac-F1 | Mic-F1 |
> | :--- | :--- | :--- | :--- | :--- |
> | SBSN | 90.67 ± 2.08 | **97.68 ± 1.12** | **72.69 ± 4.97** | **95.58 ± 2.06** |
> | SBSN-node| **90.73 ± 2.23** | 97.15 ± 1.41 | 70.26 ± 5.42 | 94.59 ± 2.58 |
>
> These more rigorous results demonstrate that the performance of SBSN and SBSN-node is, in fact, **statistically indistinguishable**. The initial observation was indeed due to random variance. This confirms that our node feature distillation module effectively preserves the predictive power of the full subgraph-based model while offering a massive gain in inference efficiency. We will update the results table in the manuscript with these more stable figures and add a note clarifying this.
>
> We are grateful for your insightful questions, which have prompted us to conduct further experiments and analysis that strengthen our paper. We believe these clarifications and new results directly address the concerns you raised. We will incorporate all these points into the revised manuscript and are confident that these changes will significantly improve its quality and clarity.

---

> > ### Author Response · Authors · 2025-08-07
> > **Request for Review Update**
> >
> > The discussion phase deadline is approaching, but the reviewer has not yet responded to the rebuttal. We kindly request the reviewer to provide feedback soon to ensure a complete and fair review process. We remain open and willing to address any new questions or concerns raised by the reviewer.

---

> ### Author Response · Authors · 2025-08-07
> **Futher Discussion and Experiments about DE**
>
> We would like to address your point with a specific focus on the third paper you mentioned, **Graph Neural Networks for Link Prediction with Subgraph Sketching.** In that work, Distance Encoding is effectively defined by the distances of each node within the subgraph to the target node pair (u, v).
>
> However, in our specific setting—link sign prediction in bipartite networks—we uniformly adopt a K=4 layer topology for subgraph extraction. This specific structure has a significant implication for DE:
>
> *   Layer 1 consists solely of the source node u.
> *   Layer 4 consists solely of the target node v.
> *   All nodes in Layer 2 are at a distance of 1 from u and 2 from v. Consequently, their distance encoding with respect to (u, v) is consistently (1, 2).
> *   Similarly, all nodes in Layer 3 are at a distance of 2 from u and 1 from v, resulting in a uniform distance encoding of (2, 1).
>
> As a result, in our framework, the sophisticated DE proposed in degenerates into a simpler "layer embedding." That is, the encoding does not provide unique information for each node but rather indicates the layer to which a node belongs.
>
> Our model design implicitly capture this layer-specific information. The message-passing module operates layer by layer, with each layer contains MLP. This means the model already learns a separate transformation for each layer. The bias matrices within each layer's MLP can effectively fulfill a role analogous to an explicit layer embedding, allowing the model to distinguish and process nodes based on their layer.
>
> To empirically validate this, we conducted experiments where we explicitly added these layer embeddings as node features. The results yielded no performance improvement. This outcome reinforces our hypothesis that our layer-wise architecture inherently and sufficiently captures this degenerated distance information.
>
> We appreciate you bringing this important line of work to our attention. In the final version of our paper, we will expand our "Related Work" section to include a discussion of the DE methods. We will also clarify why this technique reduces to layer embeddings in our specific bipartite link sign prediction setting and explain how our model implicitly addresses this.

---

> > ### Comment · Reviewer_pqpu · 2025-08-07
> >
> > Thanks for your response and hard work. I have read your rebuttal which has address most of my concerns. I have raised my score. The paper looks good to me as long as proper discussion on DE is included in future revision. Good luck.

---

### Note · Authors · 2025-08-13

**Author Final Remarks**
﻿
We thank all reviewers and the AC for their thoughtful feedback, constructive suggestions, and recognition of our work’s novelty and potential impact. We are pleased that the reviews acknowledge SBSN’s clear motivation, strong empirical results, scalability, and effective design for signed bipartite link prediction.
﻿
**Addressing Reviewer Concerns**
﻿
1. **Distance Encoding (DE):**
We conducted additional experiments incorporating DE and anchor-based initialization. While DE slightly improved early convergence, final performance remained similar, confirming that SBSN’s strength lies in dynamically learning structural context through its Topological Subgraph Extractor and Directed SignFlow Aggregator. In bipartite K=4-layer extraction, DE degenerates to layer embeddings, which our architecture already models implicitly.
﻿
2. **Distilled vs. Full Model on Bonanza:**
Extended trials (15 runs) show SBSN and SBSN-node have statistically indistinguishable performance. The initial gap was due to variance. SBSN-node thus offers substantial inference speed gains without sacrificing accuracy.
﻿
3. **Novelty Clarification:**
We explicitly contrast SBSN with SBGNN, SBGCL, and SELO:
- **Subgraph Extraction:** Path-constrained and layer-balanced sampling reduces memory and focuses on informative connecting paths.
- **SignFlow Aggregator:** Edge-centric, query-specific attention captures directional flow, unlike static relation-type weights (e.g., R-GCN).
- **Node Feature Distillation:** First application in signed bipartite link prediction for bridging subgraph-model accuracy with node-model inference efficiency.
﻿
4. **Complexity & Theorem 1 Validation:**
We moved key complexity analysis into the main text and empirically validated the optimal K from Theorem 1. Results confirm K=4 balances path coverage with avoiding over-smoothing.
﻿
5. **Reproducibility:**
All code and datasets (including Bonanza) are included in supplementary material and will be fully released in the camera-ready version.

---

### Decision · Program_Chairs · 2025-09-17

**Decision:**

Accept (poster)

**Comment:**

This paper introduces SignFlow Bipartite Subgraph Network (SBSN), a scalable and effective method for link sign prediction in large-scale signed bipartite graphs. It combines (1) path-constrained subgraph extraction, (2) a directed edge-centric message passing aggregator, and (3) a node feature distillation module for efficient inference. Several initial concerns have been well addressed in the rebuttal, and after the rebuttal, all the reviewers think this is a good paper: proposing a novel method and with sufficient empirical results.